# Long-lived hot-carrier light emission and large blue shift in formamidinium tin triiodide perovskites

Hong-Hua Fang [1], Sampson Adjokatse[1], Shuyan Shao[1], Jacky Even [2] & Maria Antonietta Loi[1]

A long-lived hot carrier population is critical in order to develop working hot carrier photovoltaic devices with efficiencies exceeding the Shockley–Queisser limit. Here, we report photoluminescence from hot-carriers with unexpectedly long lifetime (a few ns) in formamidinium tin triiodide. An unusual large blue shift of the time-integrated photoluminescence with increasing excitation power (150 meV at 24 K and 75 meV at 293 K) is displayed. On the basis of the analysis of energy-resolved and time-resolved photoluminescence, we posit that these phenomena are associated with slow hot carrier relaxation and state-filling of band edge states. These observations are both important for our understanding of lead-free hybrid perovskites and for an eventual future development of efficient lead-free perovskite photovoltaics.

[1] Zernike Institute for Advanced Materials, University of Groningen, Nijenborgh 4, 9747 AG Groningen The Netherlands. [2] Fonctions Optiques pour les Technologies de l'Information (FOTON), UMR 6082, CNRS, INSA Rennes, Université de Rennes 1, Rennes 35708, France. Correspondence and requests for materials should be addressed to M.A.L. (email: m.a.loi@rug.nl)

Hybrid organic–inorganic perovskites have attracted increasing interest for optoelectronic applications, including photovoltaics[1], lasers[2], light-emitting diodes[3], and sensors[4]. The superior intrinsic photophysical properties such as high absorption cross-section, long charge-carrier diffusion length, low trap densities and facile solution processability, make them ideal candidates for cheap and efficient solar cells[5,6]. Recently, the toxicity of the water-soluble lead-based materials has raised considerable concerns on the potential environmental impact of the large-scale use of these materials, driving the exploration of non-toxic, lead-free alternatives[7–16]. Tin-based halide perovskites are a class of alternatives under the spotlight, which have direct band gap in the near infrared. Due to their narrower optical band gap (1.3 eV for $MASnI_3$ and 1.32 eV for $CsSnI_3$)[17,18], and broader solar spectrum absorption, the tin perovskites are potentially better candidates for high performance photovoltaic devices. These effective light harvesting materials have been recently employed in single junction solar cells with reported power conversion efficiencies larger than 8%[19] and 9% in our group[20]. By combining both the tin and lead perovskites in appropriate proportions, the tin-lead alloys with band gaps as low as 1.2 eV have been synthesized. Photovoltaic devices based on alloys such as $FA_{0.75}Cs_{0.25}Sn_{0.5}Pb_{0.5}I_3$ or perovskite-perovskite 4T tandem architecture have recently been reported to display high efficiencies[13,21–23], where the low efficiency of single junction tin-only-based solar cells seems not related to the intrinsic properties of the perovskite material, but to large losses in $V_{OC}$ mainly due to trap assisted recombination[24].

It is known that when a semiconductor is excited with photons of energy higher than the band gap, hot carriers are generated and their excess energy is dissipated via phonon emission till they finally thermalize to the bottom of the band. This is a major loss channel in photovoltaics, and is partially responsible for the Shockley–Queisser efficiency limit. This limitation could in principle be circumvented if all the energy of the hot carriers could be captured, pushing in this way the efficiency up to 66%[25]. Recently, there have been intensive investigations on hot carrier relaxation in lead halide perovskites. Yang et al. and Price et al. observed a hot-phonon bottleneck in lead-iodide perovskites through femtosecond transient absorption measurements[26,27]. Zhu et al. also reported hot photoluminescence (PL) emission at room temperature in lead bromide compounds, but showed that it disappears at 77 K[28]. Yang et al., attributed the hot electrons in several hybrid perovskites to the hot-phonon bottleneck determined by the up-conversion of low-energy phonons[29]. However, the thermalization of hot carriers reported so far typically takes place on a picosecond time scale or faster[26,29,30], making it very challenging or impossible to extract them before relaxation. Slowing down the hot carrier relaxation is the holy grail of third generation photovoltaics, as it is expected to provide the possibility to extract the excess energy of the hot carriers.

Here, we demonstrate a very slow (nanosecond range) PL emission from hot carriers both at 293 and 24 K in $FASnI_3$. Unlike in lead halide perovskites, we observe a large blue shift (150 meV at 24 K and 75 meV at 293 K) of the PL peak energy as excitation density increases, which we attribute to the slow hot-carrier relaxation and consequent band filling. Insight into the hot carrier relaxation and the charge recombination mechanism in $FASnI_3$ films are obtained by studying the energy-resolved and time-resolved PL (TRPL).

## Results

### Power-dependent photoluminescence of $FASnI_3$.
Films of $FASnI_3$ were prepared on indium tin oxide (ITO) covered glass substrates by spin coating. The sample preparation is described in the methods section. The obtained $FASnI_3$ is confirmed by X-ray crystallographic data analysis (Supplementary Fig. 1). Optical absorption spectra measured at room temperature (293 K) till 24 K, and PL excitation spectra at 293 K are reported in the Supporting Information (Supplementary Fig. 2, Supplementary Fig. 3 and Supplementary Note I). The absorption onset at 293 K is localized at around 1.43 eV (865 nm)[31]. Upon cooling down, it monotonically shifts towards lower energy 1.35 eV (916 nm) at 24 K, which is in agreement with the commonly observed anomalous Varshni trend in hybrid lead-based perovskites, with a positive thermal expansion coefficient of the band gap[32]. Similar to the absorption spectra, the PL peak redshifts from 1.38 eV at 293 K to 1.24 eV at 24 K (Supplementary Fig. 4 and Supplementary Fig. 5). Both the PL and absorption spectra shift almost continuously towards low energy from 293 to 24 K. A slight change in the variation of the optical absorption spectrum which occurs at about 80 K is possibly in relation to a phase transition as is commonly observed in lead-based perovskites[32–34]. Figure 1 shows the normalized PL spectra at 293 and 24 K under various excitation densities. The PL intensity at 293 and 24 K as a function of the pump power density are summarized in Figs. 1c, f, respectively. In general, the integrated PL intensity ($I$) is proportional to $P^k$, where $P$ is the power of the exciting laser radiation. The PL intensity at room temperature increases linearly with excitation power up to about 2 μJcm$^{-2}$, suggesting that the quantum yield in this range is constant (QY ≈ 6%). Two mechanisms are usually proposed to explain the linear dependence on the excitation intensity of the PL. The first is the radiative excitonic recombination. However, this possibility is readily exempted at room temperature, because excitonic features are not observed in the absorption spectrum, suggesting a very small exciton binding energy for this material[35]. A second more suitable explanation is attributed to the recombination of photogenerated electrons with the hole density due to the unintentional doping in $FASnI_3$[31] (Supplementary Fig. 6 and Supplementary Note II). However, when the excitation density is further increased (above 2 μJcm$^{-2}$), the slope of power-dependent PL intensity becomes sublinear, which is in contrast to what has been observed for lead-based perovskite that exhibits superlinear dependence on the injected carrier density[36]. We attribute the slope variation at high excitation power in Sn-based perovskites to Auger losses (vide infra).

Interestingly, a blue shift of the emission is observed as the excitation density increases, both at 293 and 24 K. Figures 1b, e show the 2D pseudo-color plots of the normalized PL spectra. The blue-shifts of the PL peaks are as large as 75 and 150 meV at 293 and 24 K, respectively. To establish the possible origins of the band gap shift, we first consider the origins of band gap shifts in classical semiconductors and relate them to the perovskite semiconductor. The hypothesis related to exciton, such as exciton-exciton interactions or Coulomb screening of the exciton, can be readily excluded because there is no evidence for stable excitons in $FASnI_3$. Other two possible origins for $FASnI_3$ are photo-induced band filling and band gap renormalization, i.e., modification of band edge states self-energies by the screening due to photogenerated carriers. This second case can be ruled out because it gives rise to a red shift rather than the observed blue shift. In the first case, the blue shift of the optical band gap is a consequence of the dynamic free carriers filling of the densities of states, as schematically shown in Fig. 2a. Saturation in PL intensity is seen at high carrier densities (Fig. 1c; Supplementary Fig. 7). As a result, it is possible to fill band edge states, with a consequent blue shift of the transition energy and line-width broadening. Figure 2b, c show the peak shift in time-integrated PL spectra as a function of the photoexcitated carrier density. The band filling is further validated by a good fit of the carrier

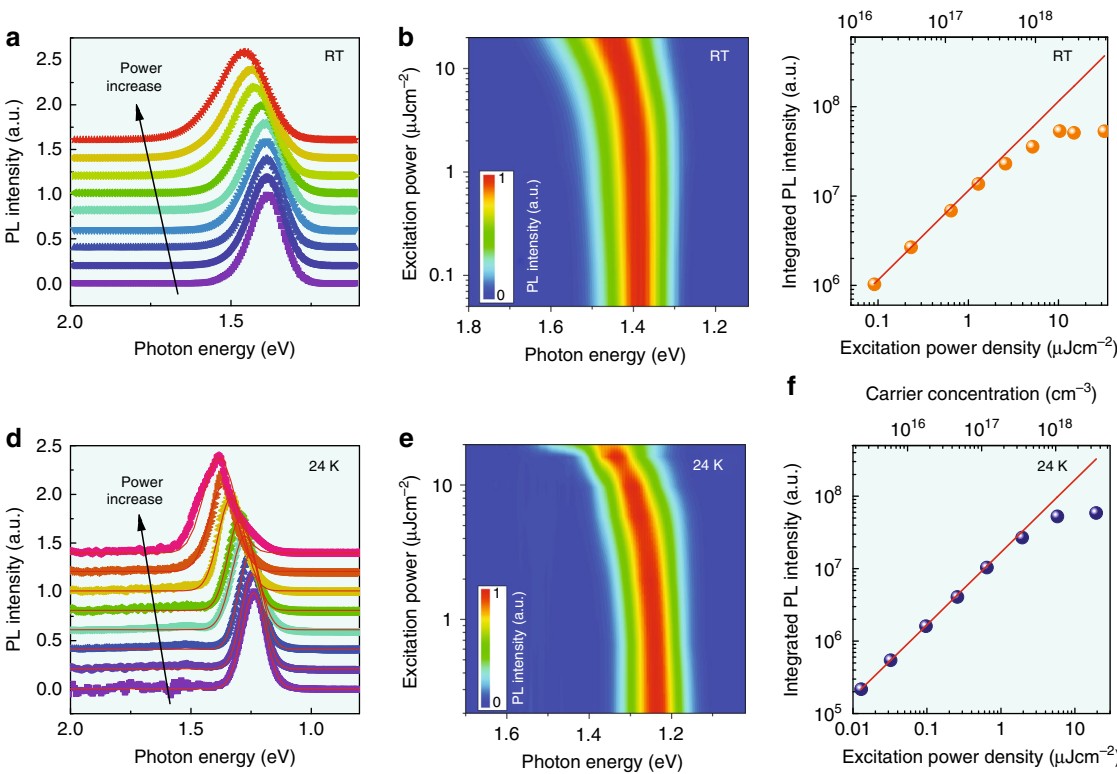

**Fig. 1** Power-dependent photoluminescence spectra of the FASnI₃ thin film. **a** Normalized PL spectra of FASnI₃ with excitation fluence increasing from 0.03 to 34 μJcm⁻² at 293 K. **b** 2D pseudo-color plot of normalized PL spectra as a function of excitation power, showing the blue shift as the excitation density increases. The emission energy shifts from 1.38 to 1.46 eV at 293 K (with similar conditions the shifts are equal to 17 meV for FAPbI₃, 6.5 meV for MAPbI₃ and 4 meV for MAPbBr₃). **c** Logarithmic plot of the integrated intensity of the emission as a function of power density—a linear dependence (red line) at low excitation fluence is observed at 293 K. **d** Power-dependent PL spectra of FASnI₃ at 24 K. **e** 2D pseudo-color plot of normalized PL spectra as a function of excitation power, PL blue shift from 1.23 to 1.39 eV at 24 K. **f** Integrated PL intensity as a function of power density at 24 K

concentration dependent emission peak shift at time = 0 (Supplementary Fig. 8, and Supplementary Note III). From the fitting, we can estimate values for the reduced effective carrier mass of 0.121 $m_0$ at 24 K, and 0.214 $m_0$ at 293 K, where $m_0$ is the mass of the electron in vacuum. These values are in line with the previously reported value of 0.2 $m_0$ in MASnI₃ obtained with a combined study of infrared reflectivity and Hall effect measurements.[34]

**Energy-resolved and time-resolved photoluminescence**. To better understand the photoexcitation process and gain an insight into the charge-carrier dynamics in FASnI₃, we performed TRPL measurements as a function of the excitation density. TRPL was measured using a streak camera and exciting the sample with laser pulses of 3.08 eV and width of about 150 fs. When the excitation density increases, the emission peak extracted immediately after excitation shifts towards high energy. Figures 3a, b show pseudo-color plots of the typical TRPL spectra of FASnI₃ thin films measured at three different excitation densities for temperature at 293 and 24 K, respectively. The PL spectra clearly show an additional signal in the higher energy side of the spectra (PL from carriers with excess energy) when the excitation density is increased (Fig. 3c, d). By normalizing the PL spectra at the low-energy side, as shown in the inset of Fig. 3c, d, it is found that the contribution of this additional signal (high-energy tail) is increased at higher excitation. Interestingly, the ratio of the emission from the high energy tail is larger at 24 K than at room temperature under similar excitation. If this additional signal at the high-energy tail of the spectrum is assumed to come from hot

carriers, then the emission from the hot carriers at 24 K is stronger than that at 293 K.

Recently, Manser and Kamat reported a blue shift in the transient bleach signal in thin films of methylammonium lead iodide[37], which they attributed to band filling by free charge carriers. However, this effect has been reported to occur only at rather short delays (5 ps after excitation)[26,37]. Moreover, it appeared to be partially compensated by the band gap renormalization effect, thus contributing little to the PL. For a comparison, we investigated the properties of FAPbI₃, MAPbBr₃, and MAPbI₃ thin films. Figure 3e–g present TRPL spectra for FASnI₃, FAPbI₃, and MAPbI₃ at similar excitation conditions, showing that the FASnI₃ thin films show much stronger band-filling effect and prolonged hot carrier emission. The power-dependent PL with similar excitation conditions for those lead-based perovskites shows much smaller peak energy shifts as a function of the excitation power (17 meV for FAPbI₃ (Supplementary Fig. 9), 6.5 meV for MAPbI₃ (Supplementary Fig. 10), and 4 meV for MAPbBr₃ (Supplementary Fig. 11 and Supplementary Fig. 12) at room temperature compared to 75 meV for FASnI₃).

**Hot carrier emission**. The normalized emission under high excitation shows a large blue shift and a high-energy front living up to 1 ns after photoexcitation. This suggests that carriers do not reach the band edge immediately, and that luminescence from carriers with excess energy is present in the time dependent spectral evolution (Supplementary Fig. 13). This is an evidence that carrier cooling is inhibited by the filled states in tin-based

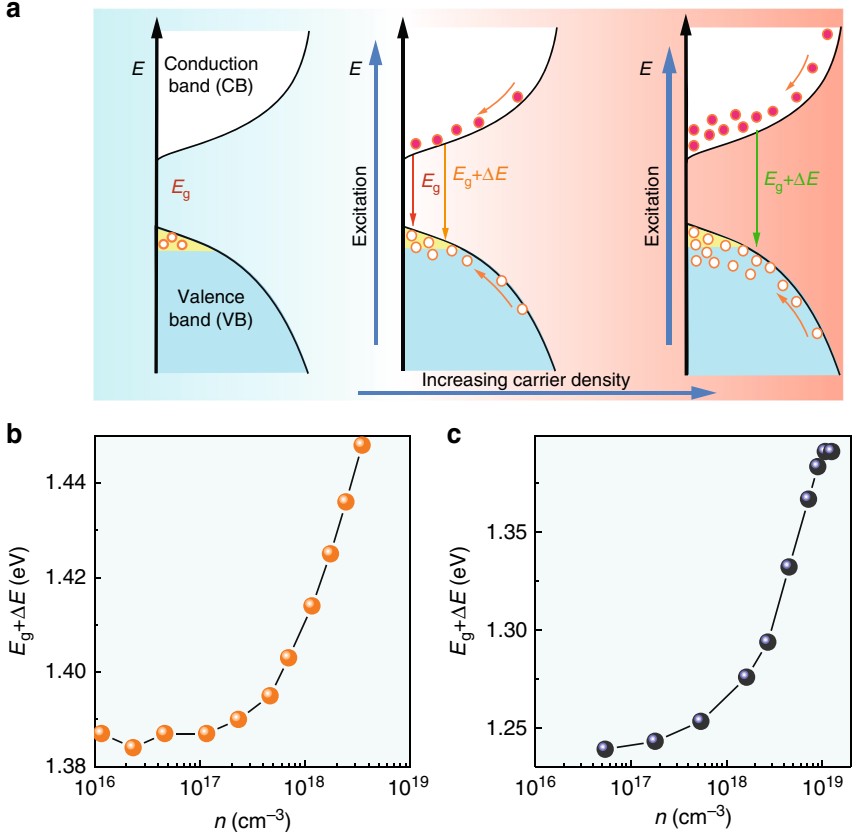

**Fig. 2** Blue shift of the photoluminescence. **a** Schematic of the mechanism for the blue shift of the PL as a result of band filling. **b** and **c** emission peak energy versus the photocarrier density $n$ at 293 and 24 K, respectively

perovskites; therefore, the dynamic band filling in FASnI$_3$ leads to a shift of the optical transitions towards high energy and to an unusually slow hot carrier relaxation time (more than 1 ns) towards the band bottom. This is especially noticeable when compared with lead-based perovskites, in which the carrier relaxation to the band edges is occurring in the picosecond regime or faster[26–28]. This unique property of FASnI$_3$ may lead to the application of Sn-based perovskites in optoelectronic devices, such as wavelength tunable light-emitting diodes and lasers.

It was recently reported that the addition of SnF$_2$ in the active layer is critical in improving the efficiency of tin-based perovskite solar cells[13,38,39]. For comparison, excess SnF$_2$ was added to the precursor solution for the fabrication of the FASnI$_3$ thin film (scanning electron microscope images and atomic force microscope images are presented in Supplementary Fig. 14 and Supplementary Fig. 15, respectively). Figure 4a shows TRPL spectra of a FASnI$_3$ film treated with SnF$_2$. The introduction of SnF$_2$ significantly increases the PL intensity (Supplementary Fig. 16) and the lifetime of the Sn-based perovskite (4 ns) in comparison to the one measured for films not treated with SnF$_2$ (0.3 ns). The longer lifetime of the PL confirms that SnF$_2$ suppresses the oxidization of Sn$^{2+}$ and thus reduces the natural p-doping of FASnI$_3$. Similar to the pristine FASnI$_3$, blue-shifted emission at increased excitation density (Supplementary Fig. 17), and a broad PL high-energy tail extending up to 1.9 eV (Supplementary Fig. 18 and Supplementary Fig. 19) is observed at early times, which is attributed to radiative recombination of hot carriers. The emission peak red shifts as time elapses, reflecting the hot carrier relaxation dynamics and the reduction of the band filling as the photo-excited carriers recombine (Supplementary Fig. 20). The PL intensity decays are shown in

Fig. 4b for emission energies of 1.65, 1.41, and 1.38 eV. The hot-carrier PL at the high-energy tail of the spectra (1.65 eV) shows an initial rapid cooling with a time constant of $t = 0.16$ ns, followed by a significantly slower relaxation process taking up to 4.5 ns. In order to obtain the carrier temperature as a function of time, the PL spectra at different delay times are extracted. The spectra have a high-energy tail that decays exponentially with energy, whose line shape is approximated to be a modified Maxwell–Boltzmann distribution[40,41]. The carrier temperature is obtained by fitting the high-energy tail of the spectra globally using PL spectra at each delay time (details of the fitting method is presented in Supplementary Note IV). In Fig. 4c, the hot-carrier temperature as a function of time is displayed. The carrier temperature rapidly decreases from ~1600 K after excitation to ~750 K at 0.2 ns after excitation. The cooling process then slows significantly down, showing a time constant of about 6 ns.

It is worth highlighting that such large blue shift is present in FASnI$_3$ also under continuous-wave excitation. The emission peak shift from around 1.4 eV under excitation of 0.027 kWcm$^{-2}$ (corresponding to 0.27 sun intensity), to 1.45 eV at 4.4 kWcm$^{-2}$ (44 suns intensity). Figure 5 shows the power-dependent PL of a FASnI$_3$ excited with a continuous-wave laser. Similar to the TRPL spectra, the spectra measured with CW excitation have a high-energy tail that decays exponentially with energy, which can be fitted with the modified Maxwell–Boltzmann distribution (red solid curves). The good fit of the high-energy tail confirms that it originates from the radiative recombination of hot carriers. Based on the normalized PL spectra, the contribution to total PL emission of the high-energy tail is estimated to be as high as 59% under 44 Suns. This strong emission from hot carriers, in continuous-wave operation, the mode of operation of solar

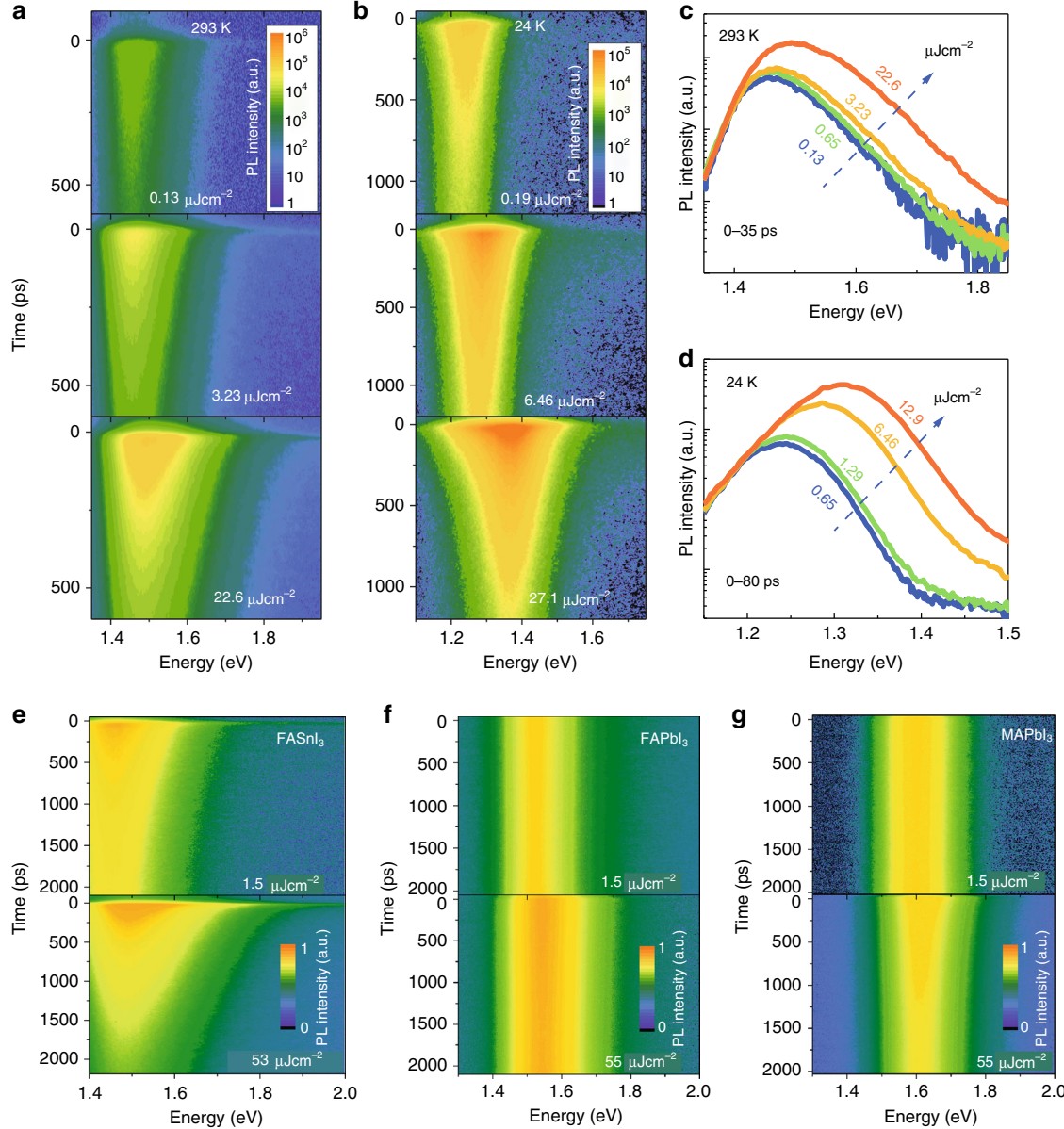

**Fig. 3** Power-dependent time-resolved photoluminescence revealing long-lived hot carriers. **a**, **b** Energy-resolved and time-resolved photoluminescence of FASnI$_3$ thin films pumped with different laser intensities at 293 and 24 K, respectively. Semi-log plot of PL spectra immediately after excitation under 3.1 eV pump excitation at different intensities at **c** 293 K and **d** 24 K. PL spectra are taken by integrating the time profile from 0 to 35 ps and 0 to 80 ps for 293 and 24 K, respectively. The corresponding excitation densities are indicated. The PL spectra have been normalized at the low-energy tail to show the hot carrier PL clearly. **e**–**g** Energy-resolved and time-resolved photoluminescence spectra of perovskite thin films at low and high excitation density for FASnI$_3$, FAPbI$_3$, and MAPbI$_3$, showing that the FASnI$_3$ thin films exhibit much stronger band-filling effect and prolonged hot carrier emission

cells, is an important first step towards harvesting these hot carriers[42,43].

## Discussion

The long-lived hot-carrier and the large contribution of them to the total PL signal raise questions about the reasons of this feature in FASnI$_3$. It is noted that FASnI$_3$ also shows very slow decaying hot PL and even larger PL blue shift at low temperature. This is in contrast to what was observed by Zhu et al.[28], which was explained by the protection afforded by molecular reorientations at room temperature, vanishing at low temperature due to the freezing of these motions. Obviously, such a mechanism can hardly explain the slow relaxation of hot carriers in FASnI$_3$, considering its persistence at low temperature. Phonon bottleneck

effect may play an important role in decreasing the hot carrier population decay rate[26,27,29,44]. Yang et al.[29], recently suggested that phonon up-conversion is essential for effective vibrational energy recycling in hybrid lead-halide perovskites, which prolongs the overall cooling period of the carrier–phonon system. Although interesting, this explanation does not provide a motivation for the difference between lead and tin compounds, as well as for the specific role of the FA cation by comparison to MA and Cs one.

Some alternative explanation could be related to a more fundamental difference between the electronic structures of tin and lead halide compounds. A first difference between lead and tin compounds might be related to the conduction band (CB). The spin–orbit coupling effect is indeed expected to be about 0.4 eV in tin compounds, that is about three times smaller than in their

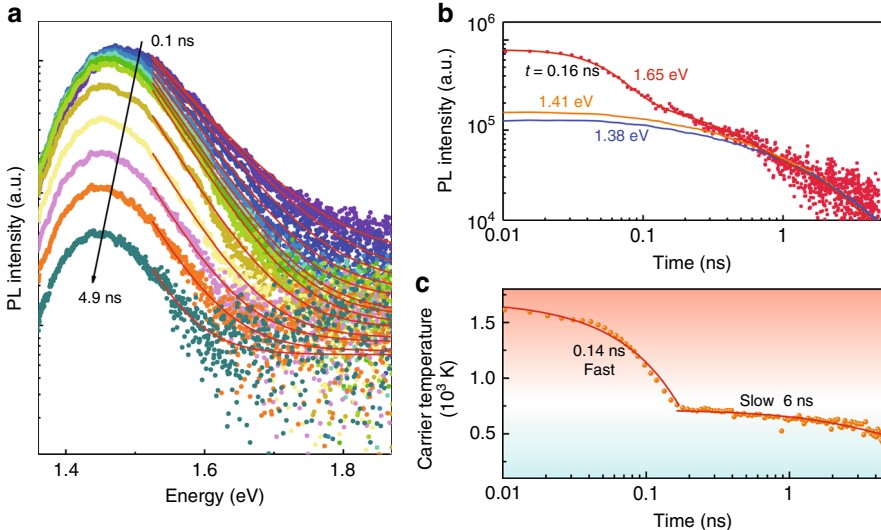

**Fig. 4** Photoluminescence from FASnI$_3$ thin film with addition of SnF$_2$. **a** Semi-log plot of PL spectra at different delay time after excitation under an excitation fluence of 28 μJcm$^{-2}$. The high-energy tail of the PL spectra is fitted with $I_{PL} \propto \alpha(E)\exp(-h\nu/kT_c)$, where $T_c$ is the carrier temperature, and $h\nu$ is photon energy. **b** PL intensity decay at 1.65, 1.41, and 1.38 eV. The PL intensity has been normalized at the tail. **c** Extracted carrier temperature from the hot-carrier distribution as a function of the delay time. The fast component is fitted to be 0.14 ns, and the slow component is around 6 ns. The carrier temperature is obtained by fitting the high-energy tail of the PL spectra at each delay time (Supplementary Note IV)

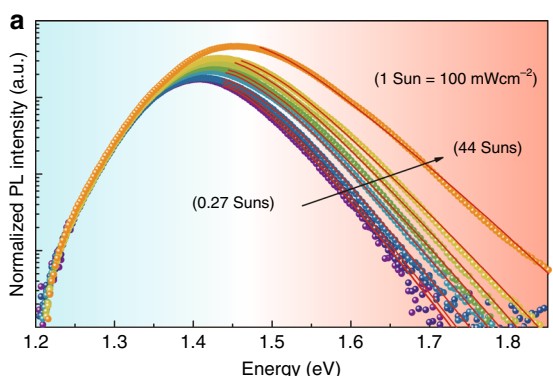

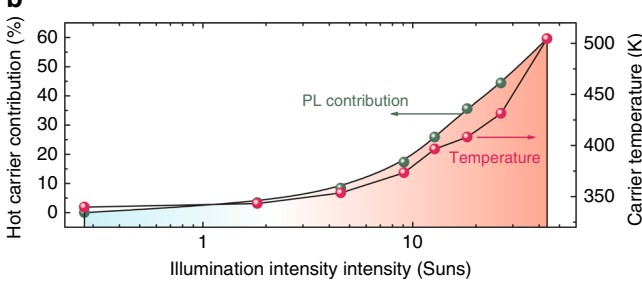

**Fig. 5** Hot carrier photoluminescence under continuous-wave excitation. **a** PL spectra of FASnI$_3$ thin film with addition of SnF$_2$ under 405 nm laser diode excitation. The fitting of the carrier temperature is reported with red solid lines. **b** Hot carrier contribution to the photoluminescence and carrier temperature as a function of illumination intensities reported in term of sun (AM 1.5, 1 sun = 100 mWcm$^{-2}$)

lead counterparts[45]. On the other hand, the high temperature phase of FASnI$_3$ was clearly refined at 293 K in an orthorhombic ferroelectric Amm2 space group exhibiting well-defined orientations of the formamidinium cations contrary to most 3D tin or lead hybrid perovskites[46]. FASnI$_3$ is thus the only example where a Rashba−Dresselhaus effect could be unambiguously predicted from the room temperature crystallographic structure[47,48]. The

importance of Rashba-like effects for the radiative lifetime of the ground state is still under debate, but the consequences of the strong modifications of the densities of states (especially at the bottom of the CB), on the carrier relaxation also certainly deserve theoretical investigations[47,48]. Several authors discussed the possibility of Rashba-like dynamical fluctuations of the CB structure[49–52], leading to either an indirect band gap behavior or a coexistence of indirect and direct band gaps.

However, in order to explain the behavior of FASnI$_3$, it is more convenient to consider the role of the valence band (VB). Using first-principles calculations, Kawai et al. propose that a reduction of the relaxation paths in the small valence electronic density of states is responsible for the slow hot-hole cooling[53]. The experimental results in the present work for p-doped FASnI$_3$ and FASnI$_3$ films with the addition of SnF$_2$ seem to indicate that the filling of the VB plays a specific role. More, as reported experimentally and theoretically[54], the unit cell of FASnI$_3$ has a larger size than the ones MASnI$_3$ and CsSnI$_3$. MA and Cs cations are thus at the origin of a chemical pressure on the inorganic lattice. On the other hand, it has been demonstrated theoretically from a tight binding representation of the halide perovskite band structure[55], that the main reduction of the electronic band gap under pressure, is related to a shift of the VB edge. Simultaneous band-gap narrowing and carrier-lifetime prolongation of organic−inorganic trihalide perovskites were demonstrated experimentally as a result of a hydrostatic pressure by Kong et al.[56]. As the pressure increases, the trap states that are already present in the subgap close to VBM become shallower. A similar phenomenon in tin compounds as a result of a chemical rather than hydrostatic pressure may be consistent with the observed differences in band gaps and recombination lifetime and lead to a deeper energetic position of the shallow defects in FASnI$_3$.

In conclusion, evidence of PL from hot-carriers with unexpectedly long lifetime (a few ns) in formamidinium tin triiodide (FASnI$_3$) are reported. The asymmetry of the PL spectrum at the high-energy edge, is accompanied by the unusually large blue shift of the time-integrated PL with increasing the excitation power (150 meV at 24 K and 75 meV at 293 K). These phenomena are associated with slow hot carrier relaxation and state-filling of band edge states. Most importantly, the hot carrier PL is

present not only upon pulsed excitation but also with CW one, which is essential for making the long-standing dream to use hot carriers in solar cells and other optoelectronic devices more realistic.

## Methods

**Materials**. The glass substrates were sequentially cleaned using detergent, demi-water, acetone, and isopropanol. The substrates were then baked in an oven at 140 °C for 10 min and treated with UV ozone for 20 min. $FASnI_3$ precursor solutions were prepared by dissolving formamidinium iodide (FAI) and tin (II) iodide ($SnI_2$) precursors with molar ratio of 1:1 in a mixed solvent of anhydrous N,N-dimethylformamide (DMF) and anhydrous dimethylsulfoxide (DMSO) in a volume ratio of 4:1 to form a 1 M solution and stirred overnight at room temperature in a nitrogen-filled glovebox. The perovskite solution was spin-coated on a glass substrate covered by ITO at 4000 r.p.m for 60 s. During the spinning, chlorobenzene (anti-solvent) was dropped on the substrate to control the morphology of the film. The samples were then annealed at 70 °C for 20 min in the nitrogen-filled glovebox. For the $SnF_2$-treated samples, the required amount of $SnF_2$ was added to the FAI/$SnI_2$ precursors to form the $SnF_2$-treated $FASnI_3$ precursor solution. For $FAPbI_3$ thin films, FAI and $PbI_2$ were dissolved in anhydrous N,N-dimethylformamide at a molar ratio of 1:1. 5 vol% hydroiodic acid was added to the FAI and $PbI_2$ solution. After spin coating at 3000 r.p.m. for 60 s on a glass substrate and then drying at 160 °C for 40 min, perovskite films were formed. For $MAPbI_3$ thin films, a precursor solution composed of an equimolar of $CH_3NH_3I$ and $PbI_2$ was spin-coated on top of cleaned glass substrate and then annealed at 100 °C in a nitrogen-filled glove box. $MAPbBr_3$ thin films were made from a precursor solution composed of an equimolar of $CH_3NH_3Br$ and $PbBr_2$, which is dissolved in a mixed solvent of DMF and DMSO in a volume ratio of 4:1.

**XRD characterization**. The X-ray diffraction was performed at ambient conditions. The X-ray data were collected using a Bruker D8 Advanced diffractometer in Bragg–Brentano geometry and operating with Cu Kα radiation source ($\lambda = 1.54$ Å) and Lynxeye detector.

**Morphology characterization**. The AFM images were obtained using the Bruker NanoScope V in the ScanAsyst mode. Scanning electron microscopy (SEM) images were obtained using the FEI Nova Nano SEM 650 with an accelerating voltage of 15 kV.

**Optical measurements**. The prepared thin films were mounted into a cryostat inside a nitrogen-filled glovebox without being exposed to air. The absorption measurements were carried out under vacuum in a cryostat (Oxford Instruments Optistat CF). A halogen lamp was used as light source and a Hamamatsu EM-CCD camera was used as the detector. PL spectra were measured by exciting the sample with 3.1 eV photons of the second harmonic of a mode-locked Ti: sapphire laser (Mira 900, Coherent). The laser power was adjusted using neutral density filters during the measurement. The excitation beam was spatially limited by an iris and focused with a 150-mm focal length lens. PL was collected into a spectrometer and recorded by a cooled array-detector (Andor, iDus 1.7 µm) or an Imaging EM CCD camera (visible sensitive) from Hamamatsu (Hamamatsu, Japan). TRPL and wavelength-resolved PL were recorded with Hamamatsu streak cameras with two different streak tubes: one measures the emission in visible range, and the other one detects PL in near-infrared region. Depending on time window used, the time resolution varied. When the streak camera is working in Synchroscan mode, the time resolution is 7 ps for 1.5 ns time window and around 10 ps for 2 ns time window. In single sweep mode, the time resolution is around 1% of the time window. For continuous wave laser excitation, a laser diode (405 nm) was used.

**Data availability**. The data that support the plots within this paper and findings of this study are available from the corresponding author upon reasonable request.

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

## Acknowledgements

We would like to thank Arjen Kamp and Theodor Zaharia for their technical support. The Groningen team would like to acknowledge funding from the European Research Council (ERC Starting Grant, Hy-SPOD, No. 306983) and the Foundation for Fundamental Research on Matter (FOM), which is part of the Netherlands Organization for Scientific Research (NWO), under the framework of the FOM Focus Group (Next Generation Organic Photovoltaics). S.A. acknowledges financial support from the NWO Graduate School funding. S.S. thanks financial support from the Marie Curie Actions— Intra-European Fellowships (IEF), SECQDSC, No. 626852.

## Author contributions

The project was conceived, planned, and co-ordinated by H.-H.F. and M.A.L. Samples are prepared by S.S. and S.A., and H.-H.F. measured PL and TRPL of the samples. S.A. performed XRD, AFM and SEM characterization. H.-H.F., J.E., and M.A.L. provided the interpretation of the results. H.-H.F. wrote the first version of the manuscript, and all authors contributed to the final version.

## Additional information

**Competing interests:** The authors declare no competing financial interests.

