## [Peer Review File · Nature Communications]

Reviewers' comments:

Reviewer #1 (Remarks to the Author):

Fang et al. report the observation of slow hot-carrier cooling in polycrystalline FASnI₃ perovskite. This is a new addition to the series of observations of long-lived hot carriers in lead halide perovskites. The results bear some similarities to what has been observed in lead halide perovskites. What is new in the current manuscript is perhaps the magnitude of blue-shift with increasing excitation density, particularly the PL results under CW excitations. While the results may eventually be publishable, the authors need to address two important problems.

1) The observed linear dependence of integrated PL yield with excitation density (Fig. 1) is intriguing. The authors' interpretations are far from satisfactory and mostly arm-waving. Do the authors have supporting evidence for attributing the results to trap-mediated recombination? What is the estimated PL quantum yield? If the authors could actually establish the dominance of trap-mediated nonradiative recombination, how would this affect the analysis of hot carrier PL?

2) The authors discussed a number of mechanisms for the observed large blue shift with increasing excitation density, but prefer the band filling model to explain the results. The authors should provide a quantitative analysis based on calculated band structure or density of states to support their model.

Reviewer #2 (Remarks to the Author):

Authors reported up to nanosecond long-lived PL from hot carrier population in formamidinium tin triiodide (FASnI₃) by observing the excitation dependent time-integrated and time-resolved PL. An unusual large blue shift of the time-integrated PL with increasing excitation power (150 meV at 24 K, and 75 meV at 293 K) was displayed; compared with the much smaller blue shift and relatively shorter lifetime of hot carriers in lead based perovskites. They then associated the significant blue shift with slowed hot carrier relaxation and state-filling of band edge states. An initial interpretation for the observed slowed cooling was proposed for Tin based perovskite. This manuscript is interesting and important for physical understanding of the slowed cooling in perovskites and potentially useful for the development of high efficiency hot carrier solar cells. The manuscript is basically well organized and clearly presented. I think this manuscript can be publishable, subject to addressing the following concerns:

1. Different from the transient absorption, the PL is the minority determined method. Authors should clarify that the observed slowed cooling represent electron and/or hole. In Fig.2a and S20, authors imply electron and hole have the same band structure and cooling rate. Authors should reconsider the carrier temperature, carrier distribution, cooling rate of electrons and holes in such a p-doping system.
2. Authors have claimed exciton is negligible in the system. Even at 24 K there is no

evident exciton peak. In interpretation authors often mention exciton-exciton or exciton-phonon coupling, should change to charge carrier.

3. Fig.1c Logarithmic plot of the integrated intensity of the emission as a function of power density - a linear dependence (red line) at low excitation fluence is observed at 293 K; significantly different from the lead based perovskite. What is the slope at 24 K? what it suggests? What is the role of defect trapping? A larger than 1 slope were observed in many other perovskites, this suggests Tin based perovskite exhibit very different carrier dynamics?

4. Various mechanisms of phonon bottleneck were previously proposed for slowed hot carrier cooling, in perovskites and other semiconductors. Authors should consider [Chung et al. Solar Energy Materials and Solar Cells, 169, 13-18/2017, phonon upconversion and Phonon Auger heating in perovskites].

5. Below 50 K, the measurement showed significantly increased Stokes shift at low excitation, is this intrinsically/partly responsible for the increased blue shift?

6. Fig.3a and 3b, 3b(24K) use a smaller scale bar (10^5) compared with 295K (10^6), this indicates the PL intensity at 24K is weaker than that at 295K?

Reviewer #3 (Remarks to the Author):

The authors reported asymmetric PL spectra obtained from FASnI₃ thin films. The high-energy edge showed lifetime as long as 0.1 ns. The authors concluded that hot carrier in FASnI₃ have unusually long lifetime (as compared to ps observed in FAPbBr₃). Since hot carrier is a very hot topic in the perovskite community, I recommend publication of this manuscript in Nature Communication. However, I hope that the authors can address the following comments.

1. The so-called "hot carriers" are not so hot. They are just the high-energy edge of the asymmetric PL peak. If these emissions are caused by the band edge state filling or other reasons, these carriers may better not be called hot carriers. PL emission above the bandgap is a well-known phenomenon in halide perovskites. Beside Rashba-like dynamical fluctuation, indirect-direct transition, laser-induced damage can also cause asymmetric PL peak and blue shift.

2. The author may measure PL excitation spectra to provide additional information for the origin for the high-energy edge in the PL spectra.

3. Two recent papers reported >17% efficiencies for mixed Sb-Pb perovskite solar cells (Nature Energy, 2, 1708 (2017); JACS 139, 1117 (2017)). The authors should consider updating the references.

Response to Reviews

Reviewers' comments:

Reviewer #1 (Remarks to the Author):

Fang et al. report the observation of slow hot-carrier cooling in polycrystalline FASnI₃ perovskite. This is a new addition to the series of observations of long-lived hot carriers in lead halide perovskites. The results bear some similarities to what has been observed in lead halide perovskites. What is new in the current manuscript is perhaps the magnitude of blue-shift with increasing excitation density, particularly the PL results under CW excitations. While the results may eventually be publishable, the authors need to address two important problems.

Our response: *We appreciate the reviewer for the positive feedback on our work. In the following we address the comments raised.*

1) The observed linear dependence of integrated PL yield with excitation density (Fig. 1) is intriguing. The authors' interpretations are far from satisfactory and mostly arm-waving. Do the authors have supporting evidence for attributing the results to trap-mediated recombination? What is the estimated PL quantum yield? If the authors could actually establish the dominance of trap-mediated nonradiative recombination, how would this affect the analysis of hot carrier PL?

Our response: *We apologize for the confusion about this point of the manuscript. The integrated PL intensity is linearly dependent on excitation density. This behavior implies that monomolecular recombination is dominant in this range of excitation density, which is in line with results reported by Milot et al. (REF.31, J. Phys. Chem. Lett. 7, 4178, 2016). This behavior is attributed to the recombination of photogenerated electrons with a constant dopant hole density, which is radiative. The doping hence introduces a radiative “pseudo-monomolecular” power dependence at low charge-carrier density. We have now added few lines of discussion in the revised manuscript.*

By comparing the PL intensity of a reference (FAPbI₃ QD, with QY of 70%), we can estimate the PL quantum yield for pristine FASnI₃ to be around 6%. We investigated also the photoluminescence spectra for the thin film treated with SnF₂, which can effectively reduce the natural p-doping of FASnI₃ and increase their photoluminescence intensity (Figure S16). In both pristine and treated samples, we observe similar blue-shift emission with increased excitation. We, therefore, speculate that the dominance of monomolecular recombination has little effect on hot carrier emission.

We now include this discussion in the supporting information.

2) The authors discussed a number of mechanisms for the observed large blue shift with increasing excitation density, but prefer the band filling model to explain the results. The authors should provide a quantitative analysis based on calculated band structure or density of states to support their model.

Our response: *In the band-filling model, the band-edge states are filled by free carriers, resulting in a shift of the Fermi level over the conduction band (CB) (or the valence band (VB)). The shift of the Fermi level increases with increasing density, resulting in an*

effective blue shift of the optical band gap. It is well known that the magnitude of the shift from free-electron theory is proportional to $n^{2/3}$, which can be modeled according to the equation: (REF.37 Nat. Photon. 8, 737, 2014)

$$\Delta E_g = \frac{\hbar^2}{2m_{eh}^*} (3\pi^2 n)^{2/3} \quad (1)$$

where n is the electron carrier concentration, m_{eh}^* is the reduced effective mass and is derived from the electron and hole effective masses, m_e^* and m_h^* , according to $\frac{1}{m_{eh}^*} = \frac{1}{m_e^*} + \frac{1}{m_h^*}$, and \hbar is the reduced Plank constant.

Figure R1. Energy- and time-resolved photoluminescence spectra of perovskite thin films for different excitation densities at 293 K (A) and 24 K (B), showing that the emission peak blue shift as excitation density increases. (C) and (D), emission peak energy at around $t = 0$ versus the photocarrier density $n^{2/3}$ at 293 K and 24 K, respectively.

To obtain a validation of the band filling interpretation, the emission peak position is plotted against the two-thirds power of the carrier concentrations ($n^{2/3}$), as shown in Figure R1. Here, we assume for $FASnI_3$ a similar effective mass for electrons and holes, and that both carriers contribute to the band filling effect. Using eq (1), a good fit of the carrier concentration dependent emission peak shift is evidently obtained both at 293 K and 24 K. The agreement with the experimental data suggests that the band filling model captures the essential physics of these phenomena in $FASnI_3$. Due to the presence of a possible Rashba effect in the structure, the evaluation of the effective masses at the DFT level is not as precise as it could be. In REF 48(ACS Nano 9, 11557, 2015) effective masses for $FASnI_3$ are similar to the ones of $MAPbI_3$ at the same level of theory. REF 47(Nat. Commun. 5, 5900, 2014) is a

more extensive study using DFT of the optoelectronic properties of FASnI₃ in various polar configurations of the lattice. Considering as an example figure 4 of REF 47, the effective masses of the conduction and valence bands can be evaluated to be about 0.17 and 0.155 respectively, yielding $m_{eh}^=0.08$. Provided that the band gap is underestimated (0.24eV) at this level of theory (a classical effect of DFT), the real effective masses are expected to be somewhat larger. This is what is reported in figure S5 of the same paper where many-body corrections have applied to get a better estimation of the band gap (1.07eV) although still underestimated. Then we conclude that effective masses in FASnI₃ should not be notably different than the ones in the other perovskites as MAPbI₃.*

We now report part of this discussion in the supplementary material.

Reviewer #2 (Remarks to the Author):

Authors reported up to nanosecond long-lived PL from hot carrier population in formamidinium tin triiodide (FASnI₃) by observing the excitation dependent time-integrated and time-resolved PL. An unusual large blue shift of the time-integrated PL with increasing excitation power (150 meV at 24 K, and 75 meV at 293 K) was displayed; compared with the much smaller blue shift and relatively shorter lifetime of hot carriers in lead based perovskites. They then associated the significant blue shift with slowed hot carrier relaxation and state-filling of band edge states. An initial interpretation for the observed slowed cooling was proposed for Tin based perovskite. This manuscript is interesting and important for physical understanding of the slowed cooling in perovskites and potentially useful for the development of high efficiency hot carrier solar cells. The manuscript is basically well organized and clearly presented. I think this manuscript can be publishable, subject to addressing the following concerns:

Our response: *We thank the reviewer for his/her positive comments and the time devoted to read our manuscript.*

1. Different from the transient absorption, the PL is the minority determined method. Authors should clarify that the observed slowed cooling represent electron and/or hole. In Fig.2a and S20, authors imply electron and hole have the same band structure and cooling rate. Authors should reconsider the carrier temperature, carrier distribution, cooling rate of electrons and holes in such a p-doping system.

Our response: *In p-doped semiconductors, the electrons can recombine with background "holes" giving rise to the monomolecular recombination displayed by the power dependence measurements. In principle it is difficult to distinguish if the hot carriers are electrons or holes with photoluminescence. But in the case of p-doped FASnI₃, hot luminescence may arise from the recombination with the background holes (vide infra), therefore we can speculate that the high energy population is likely mostly of electrons. We have revised the Figure to include the effect of the dopant in the schematic of the hot luminescence emission in FASnI₃.*

2. Authors have claimed exciton is negligible in the system. Even at 24 K there is no evident exciton peak. In interpretation authors often mention exciton-exciton or exciton-phonon coupling, should change to charge carrier.

Our response: *Indeed, we didn't observe stable exciton in the thin films. Following the suggestion, we have changed the description in the revised manuscript.*

3. Fig.1c Logarithmic plot of the integrated intensity of the emission as a function of power density - a linear dependence (red line) at low excitation fluence is observed at 293 K; significantly different from the lead based perovskite. What is the slope at 24 K? what it suggests? What is the role of defect trapping? A larger than 1 slope were observed in many other perovskites, this suggests Tin based perovskite exhibit very different carrier dynamics?

Our response: *A linear dependence (red line) at low excitation fluence is observed for both at 293 K and 24 K. This suggests that monomolecular recombination is dominant in this range, possibly deriving from the recombination of photoexcited electrons with holes coming from the unintentional p doping of the Sn based perovskite. Fig. S13a show the decay kinetics of a FASnI₃ thin film under different excitation intensities at 293 K. The PL lifetime reduces very little under the excitation intensities below $\sim 1.28 \mu\text{J}/\text{cm}^2$, this is consistent with the fact that the recombination is not dominated by traps as was found in Pb based perovskites. This*

behavior confirms that monomolecular recombination is dominant in this range of excitation density, which is line with results reported by Milot *et al.* (REF.31, *J. Phys. Chem. Lett.* 7, 4178, 2016). The discussion has been included in the revised manuscript (Supplementary).

4. Various mechanisms of phonon bottleneck were previously proposed for slowed hot carrier cooling, in perovskites and other semiconductors. Authors should consider [Chung *et al.* *Solar Energy Materials and Solar Cells*, 169, 13-18/2017, phonon upconversion and Phonon Auger heating in perovskites].

Our response: *We agree with the referee that there are various mechanisms of phonon bottleneck for slowed hot carrier cooling. We have included this reference and discussion correspondingly.*

“Phonon bottleneck effect may play an important role in decreasing the hot carrier population decay rate^{26,27,29,44}.”

44. Chung, S. *et al.* Nanosecond long excited state lifetimes observed in hafnium nitride. *Sol. Energy Mater. Sol. Cells* **169**, 13–18 (2017).”

5. Below 50 K, the measurement showed significantly increased Stokes shift at low excitation, is this intrinsically/partly responsible for the increased blue shift?

Our response: *Thanks for the comment. There is a possible phase transition at low temperature (below 70K) that is causing disorder and thus an increased Stokes shift. This may partially affect the blue shift. However, we don't think it is intrinsically responsible for the increased blue shift. For example, the stokes-shift is about 36 meV at room temperature, while the emission peak shift is 75 meV in steady state PL, and around 80 meV at t=0 in time-resolved PL spectra.*

6. Fig.3a and 3b, 3b(24K) use a smaller scale bar (10^5) compared with 295K (10^6), this indicates the PL intensity at 24K is weaker than that at 295K?

Our response: *This is due to the fact that the images are acquired with 2 different streak cameras with two different cathodes: one sensitive in the visible range, and the other one in the near infrared range. At 24 K, the photoluminescence is red-shifted, therefore the NIR streak camera was used to measure the time-resolved PL. As the photocathode radiant sensitivity of NIR streak camera is much lower than that of visible camera this results in a different level of counts.*

We have now included this information in the experimental section.

Reviewer #3 (Remarks to the Author):

The authors reported asymmetric PL spectra obtained from FASnI₃ thin films. The high-energy edge showed lifetime as long as 0.1 ns. The authors concluded that hot carrier in FASnI₃ have unusually long lifetime (as compared to ps observed in FAPbBr₃). Since hot carrier is a very hot topic in the perovskite community, I recommend publication of this manuscript in Nature Communication. However, I hope that the authors can address the following comments.

Our response: *We are grateful to the referee for supporting the publication of the paper in Nature Communications.*

1. The so-called “hot carriers” are not so hot. They are just the high-energy edge of the asymmetric PL peak. If these emissions are caused by the band edge state filling or other reasons, these carriers may better not be called hot carriers. PL emission above the bandgap is a well-known phenomenon in halide perovskites. Beside Rashba-like dynamical fluctuation, indirect-direct transition, laser-induced damage can also cause asymmetric PL peak and blue shift.

Our response: *We thank the reviewer for this reflection. The term ‘hot carrier’ is commonly referred to carriers having excess energy above the bandgap of a semiconductor or, in other words of carriers with elevated energy before the thermalisation involving interactions with the phonons [Annu. Rev. Phys. Chem. 2001. 52, 193]. The high-energy tail of the luminescence decreases exponentially in PL spectra of FASnI₃ with , suggesting an hot distribution. It is true that in the sense of energetic positions (high energy edge of the PL position), these carriers may not be considered as very energetic or very hot. Here it is rather defined in the sense of the effective temperature that defines their distribution (Figure 5b).*

2. The author may measure PL excitation spectra to provide additional information for the origin for the high-energy edge in the PL spectra.

Our response: *Following the referee’s suggestion, we measured PL excitation spectra for FASnI₃ thin films at 1.4 eV, showing a very similar behavior as the absorption. Unfortunately the measurement at the relevant energy for the hot emission 1.7eV-1.8eV is very challenging as high pump intensity is needed.*

The data have been included in the revised manuscript (Supplementary).

Figure R2. Photoluminescence excitation (PLE) spectrum of FASnI_3 thin film (detection: 1.4 eV).

3. Two recent papers reported >17% efficiencies for mixed Sb-Pb perovskite solar cells (Nature Energy, 2, 1708 (2017); JACS 139, 1117 (2017)). The authors should consider updating the references.

Our response: *As suggested, we have cited these two papers in our revised manuscript.*

“22. Zhao, D. et al. Low-bandgap mixed tin–lead iodide perovskite absorbers with long carrier lifetimes for all-perovskite tandem solar cells. *Nat. Energy* **2**, 17018 (2017).”

23. Prasanna, R. et al. Band Gap Tuning via Lattice Contraction and Octahedral Tilting in Perovskite Materials for Photovoltaics. *J. Am. Chem. Soc.* **139**, 11117–11124 (2017).”

Note: We have moved the figure (originally in Figure S12) to the main text, and combined it with Figure 3a-d to make a new Figure 3.

REVIEWERS' COMMENTS:

Reviewer #1 (Remarks to the Author):

The revision has addressed most of my questions and those of other reviewers. I recommend publication.

Reviewer #2 (Remarks to the Author):

Authors have carefully addressed reviewers' comments and accordingly revised the manuscript. The revision of the manuscript has been significantly improved. I recommend publishing this revision.

Reviewer #3 (Remarks to the Author):

The revised manuscript is acceptable for publication in Nature Communications. I have no further comments or suggestions to make.

Response to Reviews

REVIEWERS' COMMENTS:

Reviewer #1 (Remarks to the Author):

The revision has addressed most of my questions and those of other reviewers. I recommend publication.

Our response: *We appreciate the reviewer for the recommendation.*

Reviewer #2 (Remarks to the Author):

Authors have carefully addressed reviewers' comments and accordingly revised the manuscript. The revision of the manuscript has been significantly improved. I recommend publishing this revision.

Our response: *We are grateful for the recommendation from the reviewer.*

Reviewer #3 (Remarks to the Author):

The revised manuscript is acceptable for publication in Nature Communications. I have no further comments or suggestions to make.

Our response: *We would like to thank the reviewer for the positive comments and support.*